# Validation of Water Quality Monitoring Algorithms for Sentinel-2 and Sentinel-3 in Mediterranean Inland Waters with In Situ Reflectance Data



**Xavier Sòria-Perpinyà** [1,*] , **Eduardo Vicente** [2], **Patricia Urrego** [1], **Marcela Pereira-Sandoval** [1], **Carolina Tenjo** [1], **Antonio Ruíz-Verdú** [1], **Jesús Delegido** [1], **Juan Miguel Soria** [2], **Ramón Peña** [1] **and José Moreno** [1]

1 Image Processing Laboratory, Universitat de València, C/Catedrático José Beltrán Martínez, 2, 46980 València, Spain; patricia.urrego@uv.es (P.U.); marcela.pereira@uv.es (M.P.-S.); carolina.tenjo@uv.es (C.T.); antonio.ruiz@uv.es (A.R.-V.); jesus.delegido@uv.es (J.D.); ramon.pena@uv.es (R.P.); jose.moreno@uv.es (J.M.)
2 Cavanilles Institute of Biodiversity and Evolutionary Biology (ICBiBE), Universitat de València, C/Catedrático José Beltrán Martínez, 2, 46980 València, Spain; eduardo.vicente@uv.es (E.V.); juan.soria@uv.es (J.M.S.)
* Correspondence: soperja@uv.es

**Abstract:** Freshwater quality maintenance is essential for human use and ecological functions. To ensure this objective, governments establish programs for a continuous monitoring of the inland waters state. This could be possible with Sentinel-2 (S2) and Sentinel-3 (S3), two remote sensing satellites of the European Space Agency, equipped with spectral optical sensors. To determine optimal water quality algorithms applicable to their spectral bands, 36 algorithms were tested for different key variables (chlorophyll a (Chl_a), colored dissolved organic matter (CDOM), colored dissolved organic matter (TSS), phycocyanin (PC) and Secchi disk depth (SDD)). A database of 296 water-leaving reflectance spectra were used, as well as concomitant water quality measurements of Mediterranean reservoirs and lakes of Spain. Two equal data sets were used for calibration and validation. The best algorithms were recalculated using all database and used the following band relations: SDD, $R560/R700$; CDOM, $R665/R490$; PC, $R705/R665$ for S2 and $R620$, $R665$, $R709$ and $R779$ for S3, using a semi-analytical algorithm; $R700$ for TSS < 20 mg/L and $R783/R492$ (S2) or $R779/R510$ (S3) for TSS > 20 mg/L; and for Chl_a, the maximum $(R443; R492)/R560$ for Chl_a < 5 mg/m$^3$ and $R700/R665$ for Chl_a > 5 mg/m$^3$. A preliminary test with a satellite image in a well-known reservoir showed results consistent with the expected ranges and spatial patterns of the variables.

**Keywords:** water quality; Sentinel; optically complex waters; Secchi disc depth; chlorophyll a; phycocyanin; total suspended solids; CDOM; empirical models; semi-analytical model

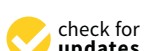

## 1. Introduction

Improving and protecting water quality, both for human needs and for the sustainability of aquatic ecosystems, has become a global priority for the 21st century [1].

The quality of inland and underground waters is growing in prominence, considering that water is an essential life element that is becoming a scarce asset due to the population rise, pollution and the consequences of global change. This is particularly critical in southern Europe, where the climatic models forecast a rise in the intensity of flooding events and drought severity for the 21st century [2]. In this scenario, the reservoirs located in basins with increased water deficit will be even more essential for supplying drinking water to the population [3]. In this context, water quality monitoring will be fundamental to assess the effect of the measures taken to keep water bodies in a good ecological state. To ensure this objective, the European Water Framework Directive (WFD) has compelled member states to establish programs for monitoring the state of inland waters. These programs require adjustment to regional and local conditions, improving sampling and

monitoring techniques for the water quality control, aiming to be as efficient and effective as possible in analysis.

Satellite remote sensing can complement in situ measurements, because, despite the fact that off in situ monitoring provides pertinent and detailed information to understand key ecosystem characteristics, it is limited to point-based representations of complex and dynamic systems and is also limited by logistics such as access, cost and timing [4]. Remote sensing allows total coverage of water ecosystems with temporal coherent data [5], although is limited to the uppermost part of the water column, is affected by bottom reflectance in shallow waters and cannot provide information in cloudy days. It has become an essential tool for the monitoring of water bodies as an assessing or warning method, which helps to obtain higher temporal and spatial resolution data. This allows optimization of the efforts for improving the information content and limit costs [6]. Continuous monitoring provides basic information for long-term monitoring records, needed to identify trends and observe climate change effects.

The new generation of Earth-Observation Sentinel satellites, belonging to the Copernicus European Union Earth Observation Program, includes two important missions for water monitoring: Sentinel-2 (S2), started in 2015, and Sentinel-3 (S3), in 2016. Both missions are formed by a constellation of two satellites to achieve a higher pass frequency.

Sentinel-2 satellites (S2-A and B, launched in 2015 and 2017, respectively), have the MultiSpectral Instrument (MSI) onboard, with spatial resolutions of 10, 20 and 60 m and a revisit time of 5 days at the equator [7]. Ten-meter resolution has not been available in open-access images until recently, making the S2 an excellent tool for intensifying studies on inland water bodies.

However, being primarily designed for land studies, the broad bands of S2 (Figure 1) make it difficult to find the specific features (peaks, shoulders, troughs) caused by the water optically active constituents (OAC) in the water-leaving reflectance. Sentinel-3 satellites (S3-A and B, launched in 2016 and 2018, respectively), which are primarily designed for water studies, have the Ocean and Land Color Instrument (OLCI) onboard, with narrower bands (Figure 1) specifically positioned for some of these OAC. The S3 OLCI images have a much coarser spatial resolution than S2 (300 m at nadir), but a revisit time of less than 2 days at the equator [8].

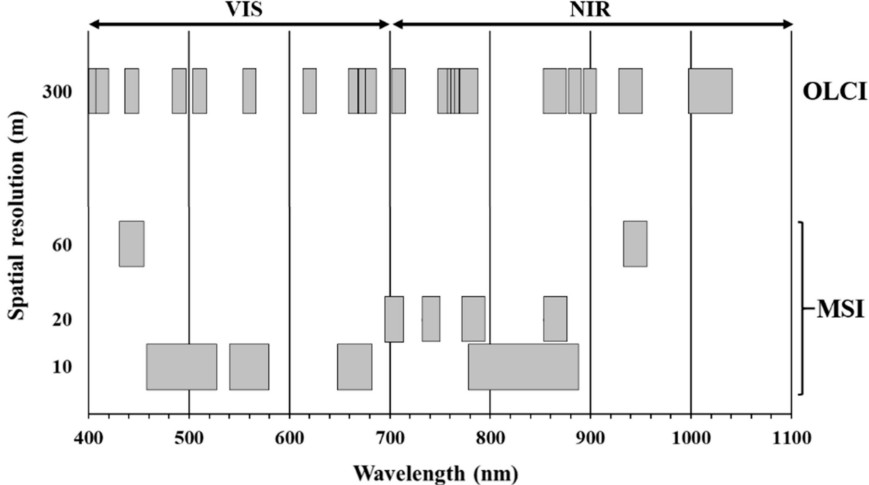

**Figure 1.** Spectral bands and spatial resolution of MultiSpectral Instrument (MSI) and Ocean and Land Color Instrument (OLCI) sensors.

Using remote sensing, water quality and ecological state can be determined through its optical properties. Optical variables often derived quantitatively from remotely sensed data include phytoplankton pigments concentration like chlorophyll a (Chl_a) or phycocyanin

(PC), water transparency like the Secchi disk depth (SDD), total suspended solids (TSS) and colored dissolved organic matter (CDOM).

Increase of chlorophyll and algal biomass are symptomatic signs of alteration in nutrient parsimony (efficiency in recycling scarce materials) and food chain structure related to eutrophication, and, as a result, are included in the normative definitions of ecological status classes for all the surface water categories [9]. Chl_a is commonly used as a proxy for phytoplankton biomass and is the most retrieved variable in water remote sensing studies. Currently, the most widely used algorithm in our target sensors is the three-band model of Dall'Olmo et al. [10], using 750, 710 and 670 nm bands [11–13] and a band ratio between 700 and 665 nm [12,14–16].

Among all the taxonomic classes of phytoplankton, examples, such as Cyanobacteria, are characteristic, because they present greater abundances in worse state water bodies [17]. Some species of Cyanobacteria can produce toxins, affecting recreational uses and drinking water supplies [18]. The early detection of these toxic substances is of interest for the assessment of risks that may affect human health [19]. Remote sensing could be an optimal tool for the detection of Cyanobacteria if the adequate bands are available. The PC, which is pigment characteristic of Cyanobacteria, has a maximum absorption at 620 nm that can be detected with S3, but not with S2. Other bands in the VIS-NIR spectral region have already been used to detect Cyanobacteria with these two sensors. From the five consulted studies, three used a band ratio between a band within 700–750 nm and another band within 600–700 nm [20–22]; one four-band model used $R620$, $R560$, $R709$ and $R754$ bands [23]; and one semi-analytical approach used $R620$, $R665$, $R709$ and $R779$ bands [24].

Water transparency is a key variable for monitoring, because the amount of light penetrating thorough the water column restricts the rate at which benthic algae, phytoplankton and macrophytes can assimilate energy through photosynthesis [25]. The most widely used and conventional method for measuring water column transparency is SDD. This is an important variable, because it can relate to the euphotic zone, which is the layer of water that has a depth where only 1% of incident light arrives [26], thus concentrating the vast majority of photosynthetic activity. Transparency is correlated with the red band, and when water clarity decreases, the brightness in the red spectral region usually increases due to particulate scattering [27]. Recent studies in S2 and S3 used different band ratios for SDD retrieval: either the $R490/R709$ ratio [13,28], the $R560/R709$ ratio [28,29] or the $R490/R560$ ratio [30].

The transparency of the water is strongly influenced by TSS, which is related directly to many other variables (e.g., turbidity, Secchi depth, water color), and, because of that, it is an important variable to consider in water management [25]. As discussed in Ruddick et al. [31], the bands more appropriate for TSS retrieval depend on the concentration: for high TSS concentrations, the best results are obtained with longer wavelengths (700 nm and beyond), whereas, for low concentrations, it is more appropriate to use shorter wavelengths (e.g., green waveband). This could explain the great variability of the obtained band combinations in the studies carried out so far with S2 and S3. In fact, six different band combinations in the five revised papers for TSS retrieval were recorded.

Inland water CDOM is mainly composed of humic and fulvic acids from organic matter degradation [26]. It is being studied increasingly with remote sensing, since its estimation allows a better understanding of the carbon cycle [32] as a part of the dissolved organic carbon budget. It is also important for ecosystem metabolism [1] and in the formation of cancerogenic chlorinated byproducts during water supply treatments [33,34]. CDOM absorption is highest in the blue part of the spectrum but the low reflectance and largest atmospheric correction errors in this spectrum region show that the green–red ratio is the most suitable for mapping lake CDOM [35,36]. For this reason, a green–red ratio was applied in four works [15,37–39] using S2 and S3, and another three applied a ratio using blue and red bands [27,39,40].

With S2 and S3 sensor characteristics, the greater spectral resolution of the OLCI sensor should provide more accurate algorithms, with bands closer to the optimal wavelengths

for the retrieval of water quality variables, while the greater spatial resolution of the MSI sensor allows to study a larger number of water bodies. The application of S3 algorithms can help us to check the coherence of S2 algorithms observing the variables estimations and provide information more frequently. Therefore, the aim of this manuscript is to find combined algorithms for S2 and S3 that give us correlated information, allowing the continuous monitoring of the spatial and temporal variation of key variables to determine the inland water ecological status.

## 2. Materials and Methods

### 2.1. Study Area

A total of 296 sampling points in 2 lakes and 50 reservoirs were analyzed, situated in several Spanish watersheds (Figure 2). The water bodies had total volumes ranging from $6 \times 10^6$ m$^3$ (Regajo reservoir) to $3219 \times 10^6$ m$^3$ (La Serena reservoir) and a maximum depth between 2 m in Albufera lake and 202 m in Almendra reservoir. The climate variability greatly influences the water quality, and the sample sites correspond to five different climate types of Köppen–Geiger classification for the 1981–2010 period: Cfa, Csa, Cfb, Csb and Bsk. Reservoir water levels vary broadly throughout the year and are therefore highly dependent on upstream rainfall and water temperature, meaning that residence time oscillates from 0.08 yr (Cortes reservoir) to 3.59 yr (La Serena reservoir). In addition, other nonclimate factors, such as altitude, vary from 1 m.a.s.l. in Albufera lake to 1100 m.a.s.l. in Riaño reservoir. An overview of sampling sites and their characteristics is given in Appendix A.

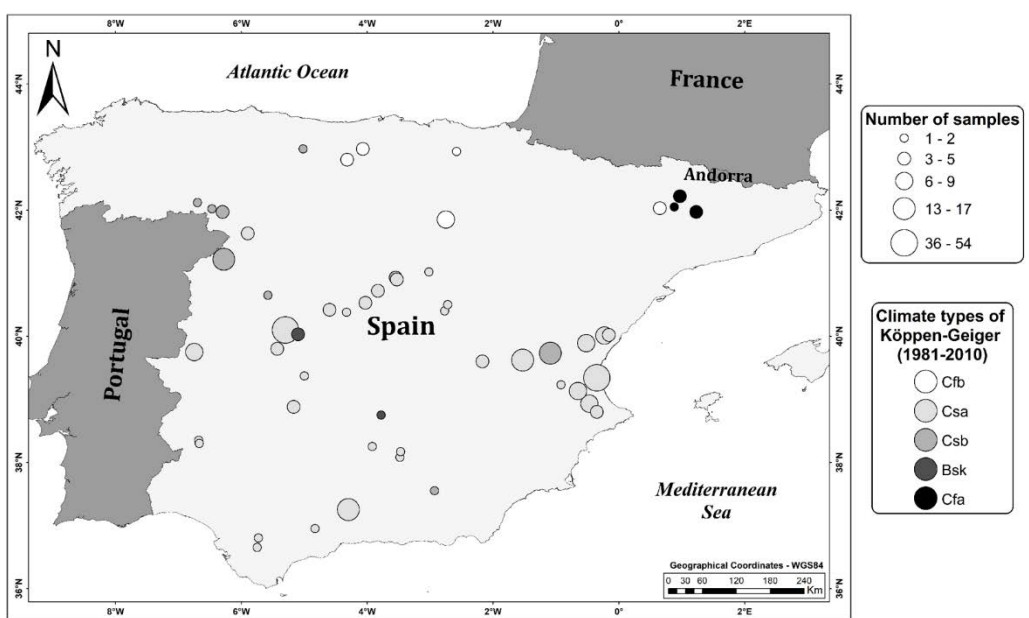

**Figure 2.** Location of sampled lakes and reservoirs. Symbol size reflects the number of samples collected at each site, and grayscale reflects the climate type.

### 2.2. Field Data Collection and Laboratory Measurements

This work is based on two data sets of previous projects, the project of the Centre for Hydrographic Studies (CEDEX data set) carried out across six years between September 2001 and July 2007 and the Ecological Status of Aquatic Systems with Sentinel Satellites project (ESAQS data set) performed between 2017 and 2018. We combined both in a single database. All sampling points were georeferenced, and the samples were taken and filtrated in the field (CEDEX) or on the same day at the laboratory (ESAQS). The availability of the different data types for each of the water bodies is given in Appendix A.

SDD and radiometric measurements were carried out in the field in both projects. In situ fluorometers were also used in both projects, in the CEDEX project for all CDOM measurements and for some PC measurements, and in the ESAQS project for all PC measurements.

SDD was measured using a disk (20 cm in diameter) with alternating black and white quadrants, which was submerged vertically until it was no longer visible. SDD consists of measuring the mean value of the point at which the disc completely disappears and the depth at which it reappears [41]. SDD was used as the sample depth in ESAQS project, but in CEDEX project, the sampling depth was that corresponding to the first optical depth ($Z_{90}$), from which 90% of the remotely sensed light comes from [42]. The first optical depth was obtained directly from downward irradiance attenuation profiles or by means of an empirical relationship with SDD. PVC tubes were used to take samples from the surface down to the calculated depth, because integrated samples are more representative and avoid missing possible phytoplankton peaks. In turbid lakes (SDD < 1 m), samples were taken directly from the surface layer.

In situ CDOM was measured using a Seapoint UV fluorometer (Seapoint Sensors Inc., Brentwood, NH, USA), and due to a factory calibration against quinine sulphate, CDOM is expressed in micrograms per liter of quinine sulphate equivalents. In situ PC was determined using a C3 Submersible Fluorometer (Turner Design Instruments; San Jose, CA, USA) in ESAQS project and a Minitracka II PA Fluorometer Model HB202 (Chelsea Instruments Ltd., Surrey, UK) in CEDEX project. The calibration of fluorescence intensity was performed by means of a calibration curve made with a standard phycocyanin extract from *Spirulina* sp. (Sigma–Aldrich Chemicals). For the radiometric measurements, three field spectroradiometers were used, one in CEDEX project and two in ESAQS project. Spectroradiometers characteristics are given in Table 1, and Appendix A shows which ones were used at each sampling location. The water-leaving radiance was obtained by measuring above-surface water spectroradiometry, according to the methodology described by Mobley [43].

**Table 1.** Details on field spectroradiometers used to collect water-leaving radiance.

| Dataset | CEDEX | | ESAQS |
|---|---|---|---|
| Instrument | ASD-FR | ASD FieldSpec® HandHeld 2 | Ocean Optics HR4000-UV-NIR |
| Manufacturer | Analytical Spectral Devices, Inc.; Boulder, CO, USA | | Ocean Optics; Largo, FL, USA |
| Acceptance angle | 8° | 8° | 8° |
| Spectral interval | 1.4 nm | 1 nm | 0.2 nm |
| Spectral range | 350–1000 nm | 325–1075 nm | 200–1100 nm |

Once the in situ remote sensing reflectance spectra was calculated according to the methodology described by Mobley [43], it was convoluted to the S2–MSI and S3–OLCI spectral bands, using the Sentinel-2 Spectral Response Functions (S2–SRF) v3.0 [44] and the Sentinel-3 Spectral Response Functions (S3–SRF) v2.0 [45].

Laboratory analyses were carried out in both projects for Chl_a and TSS. In addition, CDOM measurements in ESAQS project and the PC in CEDEX project were analyzed in the laboratory.

TSS were analyzed using the gravimetric method (ISO-11923-1997). Chl_a and CDOM in ESAQS project were measured in the laboratory using spectrophotometric methods. Chl_a samples were filtered through 0.4–0.6-µm GF/F glass fiber filters, extracted using standard methods [46] and calculated with Jeffrey and Humphrey [47] methods. The fitting function determined by Sòria-Perpinyà et al. [48] between spectrophotometric and HPLC Chl_a values was used to adjust our Chl_a results. The water filtered for Chl_a extraction was used to measure CDOM. The UV absorption curve (200–900 nm) was measured with a spectrophotometer using a 1 cm quartz cuvette and using a calibration curve with quinine

sulphate to obtain organic matter values expressed in micrograms per liter of quinine sulphate equivalents at UV absorption wavelength of 250 nm [49]. The methodology used to measure Chl_a by HPLC and to PC quantification for CEDEX samples was described by Simis et al. [24].

To assess the strength of the association between water quality variables, nonparametric Spearman correlation coefficients were calculated because the data did not have a normal distribution.

### 2.3. Algorithms Retrieval

To find the best algorithms for estimating water quality variables a comparison of the performance of different water quality product algorithms was done (Table 2). Our bibliographic research was centered in SMI and OLCI sensors from 2015, when S2 was launched, until 2020. Except for two variables, PC and TSS, we also adapted algorithms of other sensors due to the limited number of publications for these variables.

In total 36 bands combinations were tested: 14 to derive Chl_a values, 6 for CDOM, 6 for TSS, 6 for PC and 4 for SD. Bands combinations are given in Table 2.

The process for the calibration and validation of the algorithms was developed in two steps. For the first step, the data were equally divided into two data sets, one for calibration and another for validation.

Once we had two data sets, the algorithms were calibrated with a regression between the field data, using raw or transformed values, and bands combinations using S2–MSI and S3–OLCI convoluted spectral bands, finding the best fitting function (linear, potential, exponential or polynomial). The algorithms were validated by four error statistics using observed and estimated data: coefficient of determination ($R^2$), root–mean–squared error (*RMSE*), relative root–mean–squared error (*RRMSE*) and bias. The coefficient of determination was calculated by adjusting a linear regression between field and estimated data.

$$RMSE = \sqrt{\frac{\sum_{i=1}^{N}\left(x_i^{estimated} - x_i^{measured}\right)^2}{N}} \tag{1}$$

$$RRMSE = \frac{RMSE}{\sum_{i=1}^{N} x_i^{measured}/N} \times 100 \tag{2}$$

$$Bias = \frac{\sum_{i=1}^{n}\left(x_i^{estimated} - x_i^{measured}\right)}{N} \tag{3}$$

In the second step, the best performing algorithms for each variable were recalculated using the entire database with the aim to obtain a more robust and accurate algorithm.

**Table 2.** Band math and original specified wavelengths in nm for each algorithm used to retrieve the water quality parameters. SEE (Standard error of the estimates), MAE (Mean absolute error).

| Reference | Sensor | Atmospheric Correction | Variables | Bands Relation | N | $R^2$ | RMSE | Data Range |
|---|---|---|---|---|---|---|---|---|
| [11] | S2-MSI | 6S | Chl_a | $\left(\frac{1}{R670} - \frac{1}{R710}\right) \times R750$ | 8 | 0.78 | 5.34 | 2.89–22.83 mg/m$^3$ |
| | | | | $\frac{R705 - R665}{705 - 665}$ | 6 | 0.93 | 12.09 | 19.51–87.63 mg/m$^3$ |
| | | | | $R740/R560$ | 7 | 0.98 | 58.90 | 75.89–938.97 mg/m$^3$ |
| [12] | S3-OLCI | Bright Pixel Atmospheric Correction | Chl_a | $R709/R665$ | 15 | 0.95 | 6.53 | 1.81–96.41 mg/m$^3$ |
| | | | | $\left(\frac{1}{R665} - \frac{1}{R709}\right) \times R754$ | | 0.95 | 7 | |
| [13] | S2-MSI | Simulated water leaving radiance (Hydrolight) Sen2cor | Chl_a | $R740 \times \left(\frac{1}{R665} - \frac{1}{R705}\right)$ | 392 | 0.99 | 23 (MAE) | 10–169 mg/m$^3$ |
| | | | | $\log_{10}$ [max. $(R443; R490)/R560$] | 392 | 0.97 | 0.89 (MAE) | <10 mg/m$^3$ |
| | | | SD | $R490/R705$ | 60 | 0.68 | 0.88 (MAE) | 0.25–10 m |
| [14] | S2-MSI | Empirical line method | Chl_a | $R709/R665$ | 56 | 0.76 | 4.39 | 7.84–60.95 mg/m$^3$ |
| [15] | S2-MSI | Sen2Cor | CDOM | $R560/R705$ | 41 | 0.88 | 0.73 | 0.11–8.46 m$^{-1}$ a (440) |
| | | | Chl_a | $R705/R665$ | | 0.49 | 9.97 | 1.62–51.68 mg/m$^3$ |
| [16] | S2-MSI S3-OLCI | In situ reflectance | Chl_a | $R705/R665$ | 72 | 0.78 | 10.44 | 0.97–117.24 mg/m$^3$ |
| | | | | $R709/R665$ | | 0.76 | 10.77 | |
| [20] | Dron | | PC | $R709/R620$ | 92 | 0.95 | – | 0.43–13.07 mg/m$^3$ |
| [21] | S2-MSI | Sen2cor | PC | $R740/R665$ | 21 | 0.84 | 141 | 10–1287 mg/m$^3$ |
| [22] | S2-MSI S3-OLCI | In situ reflectance | PC | $R740 - R665$ | 29 | 0.70 | 4.82 | 0–23 Relative Fluorescence Units |
| | | | | $R707/R679$ | 9 | 0.86 | 1.45 | |
| [23] | S3-OLCI | In situ reflectance | PC | $\left(\frac{1}{R620} - \frac{0.4}{R560} - \frac{0.6}{R709}\right) \times R754$ | 216 | 0.69 | 27.69 | 0.33–317.74 mg/m$^3$ |
| [24] | MERIS | In situ reflectance | PC | $R620; R665; R709; R779$ | 373 | 0.74 | – | 0.4–1000 mg/m$^3$ |
| [27] * | S2-MSI S3-OLCI | C2RCC C2X TOA | Chl_a | $R665/R709$ | 49 | 0.7 | 8.9 | 18.9 (115.7) ** mg/m$^3$ |
| | | | TSS | $R700$ | | 0.7 | 3.5 | 8.9 (52.1) ** mg/L |
| | | | CDOM | $R665/R490$ | | 0.6 | 0.8 | 5.5 (11.7) ** m$^{-1}$ a (400) |
| | | | SD | $R490/R709; R560/R709$ | | 0.8 | 0.4 | 0.9 (6.27)** m |
| [28] | S2-MSI | Sen2cor | SD | $R560/R704$ | 79 | 0.67 | 0.06 | 0.19–0.62 m |
| [29] | S2-MSI | Polymer | SD | $R490/R560$ | 82 | 0.8 | 1.4 | 0.5–10.5 m |

**Table 2.** *Cont.*

| Reference | Sensor | Atmospheric Correction | Variables | Bands Relation | N | $R^2$ | *RMSE* | Data Range |
|---|---|---|---|---|---|---|---|---|
| [37] | S2-MSI | Sen2Cor | CDOM | $R560/R665$ | 41 | 0.65 | 1.71 | 0.14–12.24 m$^{-1}$ a(420) |
| [38] | S2-MSI | TOA | Chl_a CDOM | $R705 - \frac{R665+R740}{2}$ $R560/R665$ | 23 | 0.83 0.72 | – – | 3.6–72.9 mg/m$^3$ 1.77–15.8 mg/L a(380) |
| [39] | S2-MSI S3-OLCI | *In situ* reflectance | CDOM | $Ln(R490/R740)$ $Ln(R510/R753)$ | 32 | 0.86 0.86 | 0.44 0.44 | 0.71–4.3 m$^{-1}$ a(440) |
| [40] | S2-MSI S3-OLCI | Simulated water leaving radiance (Hydrolight) | CDOM | $R705/R490$ | – | 0.97 0.96 | 0.17 0.19 | 1–86 m$^{-1}$ a(400) |
| [50] | S2-MSI | ACOLITE POLYMER (sun-glint) | TSS | $R664$ $R865$ | 48 | 0.63 0.95 | 25.06% 10.28% | <150 mg/L >150 mg/L |
| [51] | S2-MSI | Empirical line method | Chl_a | $R560/R665$ | 30 | 0.68 | 0.14 (SEE) | 1.58–6.00 mg/m$^3$ |
| [52] | S3-OLCI | UV-AC | TSS | $R779/R510$ | 50 | 0.91 | 19.29 | 33.88–695.24 mg/L |
| [53] | Tiangong 2 MWI | UV-AC | TSS | NISSI $= R820 - R'820$ $R'820 = R750 + (R980 - R750)$ $\times (820 - 750)/(980 - 750)$ | 92 20 | 0.85 0.76 | 22.7 | 1.78–330.43 mg/L 23.12–208.89 mg/L |
| [54] | MODIS | Land-based atmospheric correction method | TSS | $R645/R555$ | 92 | 0.88 | 34.20% | 1–300 mg/L |
| [55] | OLCI | In situ reflectance | Chl_a | $\log_{10} [\text{max.} (R443; R490; R510)/R560]$ | 2720 | 0.86 | – | 0.012 to 77.9 mg/m$^3$ |

* Most repeated band ratios among those selected for each optical water type; ** Median, and range in parenthesis.

## 3. Results

### 3.1. Measured Data

Getting a wide range of values is very important for achieving a robust algorithm. Descriptive statistics for each variable are given in Table 3. These statistics show that our data set covers a wide range of ecological states for water bodies.

**Table 3.** Descriptive statistics of the measured water parameters: Secchi disc depth (SDD), coloured dissolved organic matter (CDOM), total suspended sediment (TSS), chlorophyll a (Chl_a), and phycocyanin (PC).

|  | *n* | Min. | Max. | Mean | Median | SD |
|---|---|---|---|---|---|---|
| SDD (m) | 271 | 0.10 | 11.50 | 2.47 | 1.76 | 2.26 |
| CDOM (µg/L QSE) | 222 | 0.03 | 17.14 | 1.90 | 1.60 | 1.66 |
| TSS (mg/L) | 92 | 0.67 | 78.82 | 9.03 | 3.01 | 15.92 |
| Chl_a (mg/m$^3$) | 254 | 0.53 | 704.97 | 53.26 | 10.28 | 108.87 |
| PC (mg/m$^3$) | 170 | 0.10 | 1040.00 | 106.37 | 17.27 | 194.60 |

Figure 3 summarizes the information to understand the distribution of the samples over the ranges of the variables. All variables showed a skewed distribution, with the average in the lower values, and, for the pigment data, a few values up to three orders of magnitude higher.

The cross correlations between variables are given in Table 4. The SDD is inversely related to the other four variables since all produce a reduction in transparency with increasing concentrations. CDOM shows a high correlation with Chl_a ($R^2 = 0.68$, $p < 0.001$), indicating a predominant autochthonous origin of organic matter in this data set. TSS and Chl_a are well-correlated ($R^2 = 0.65$). This indicates that, in most cases, TSS is mainly composed of phytoplankton and not of suspended minerals. Chl_a and PC are strongly related ($R^2 = 0.85$, $p < 0.001$), indicating the importance of the Cyanobacterial group in the phytoplankton composition of our data set.

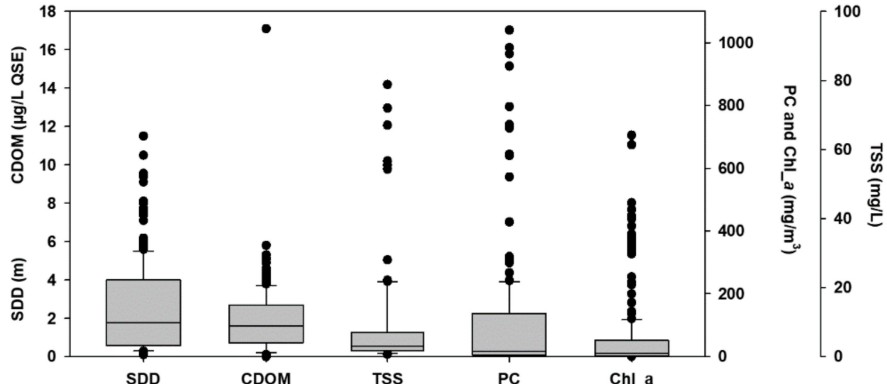

**Figure 3.** Boxplot of the values range for the water quality parameters. The box bounds the interquartile range (IQR; 25–75 percentile), the horizontal line inside the box indicates the median and whiskers (error bars) indicate the 90th above and 10th below percentiles. Dots indicate the outliers.

**Table 4.** Spearman correlation coefficients, probability value and data points (in parenthesis).

| $p < 0.001$ | CDOM | TSS | Chl_a | PC |
|---|---|---|---|---|
| SDD | −0.588 (204) | −0.831 (91) | −0.845 (251) | −0.815 (157) |
| CDOM |  | 0.551 (55) | 0.680 (202) | 0.376 (159) |
| TSS |  |  | 0.649 (92) | 0.734 (21) |
| Chl_a |  |  |  | 0.848 (160) |

### 3.2. Algorithms Retrieval

Calibration results, relating measured values of five studied variables and different band combinations, and error statistics results, relating measured and estimated values, are summarized together in the Tables 5–9.

**Table 5.** Calibration and validation results for Secchi disk depth (SDD). Only results with a calibration with $R^2 > 0.4$ are shown. All are linear regressions. *RMSE* (m); *RRMSE* (%); bias (m). $R^2$: coefficient of determination; *RMSE*: root–mean–squared error; *RRMSE*: relative root–mean–squared error.

| | Sensor | Bands Relation | Calibration | | Validation | | | | |
|---|---|---|---|---|---|---|---|---|---|
| | | | n | $R^2$ | n | $R^2$ | *RMSE* | *RRMSE* | Bias |
| SDD | S2 | $R492/R705$ | 134 | 0.65 | 131 | 0.71 | 1.30 | 55.19 | 0.35 |
| | S3 | $R490/R709$ | | 0.64 | | 0.69 | 1.36 | 57.65 | 0.36 |
| | S2 | $R560/R705$ | 135 | 0.63 | 131 | 0.77 | 1.03 | 43.64 | 0.31 |
| | S3 | $R560/R709$ | | 0.62 | | 0.77 | 1.05 | 44.46 | 0.32 |
| | S2 | $R492/R560$ | 135 | 0.60 | 131 | 0.65 | 1.21 | 51.22 | 0.10 |
| | S3 | $R490/R560$ | | 0.60 | | 0.65 | 1.19 | 50.95 | 0.13 |

**Table 6.** Calibration and validation results for colored dissolved organic matter (CDOM). Only results with a calibration with $R^2 > 0.4$ are shown. Regressions: linear (l.); potential (p.); exponential (e.). *RMSE* (µg/L QSE); *RRMSE* (%); bias (µg/L QSE).

| | Sensor | Bands Relation | Calibration | | Validation | | | | |
|---|---|---|---|---|---|---|---|---|---|
| | | | n | $R^2$ | n | $R^2$ | *RMSE* | *RRMSE* | bias |
| CDOM | S2 | Ln($R492/R740$) | 108 | 0.46 (l.) | 109 | 0.48 | 0.93 | 50.46 | 0.04 |
| | S3 | Ln($R510/R753$) | | 0.45 (l.) | | 0.47 | 0.93 | 50.85 | 0.05 |
| | S2 | $R560/R665$ | 107 | 0.51 (p.) | 107 | 0.51 | 0.95 | 52.34 | 0.29 |
| | S3 | $R560/R665$ | | 0.50 (p.) | | 0.48 | 0.98 | 53.96 | 0.29 |
| | S2 | $R560/R705$ | 108 | 0.51 (e.) | 110 | 0.55 | 0.91 | 49.88 | 0.30 |
| | S3 | $R560/R709$ | | 0.50 (e.) | | 0.55 | 0.92 | 50.15 | 0.30 |
| | S2 | $R665/R492$ | 108 | 0.49 (l.) | 110 | 0.53 | 0.88 | 47.94 | 0.03 |
| | S3 | $R665/R490$ | | 0.47 (l.) | | 0.51 | 0.90 | 48.90 | 0.03 |
| | S2 | $R705/R492$ | 108 | 0.48 (p.) | 108 | 0.48 | 1.03 | 56.24 | 0.26 |
| | S3 | $R709/R490$ | | 0.47 (p.) | | 0.45 | 1.09 | 59.14 | 0.25 |

**Table 7.** Calibration and validation results for TSS. Only results with a calibration with $R^2 > 0.4$ are shown. Regressions: linear (l.); potential (p.); exponential (e.). *RMSE* (mg/L); *RRMSE* (%); bias (mg/L).

| | Sensor | Bands Relation | Calibration | | Validation | | | | |
|---|---|---|---|---|---|---|---|---|---|
| | | | n | $R^2$ | n | $R^2$ | *RMSE* | *RRMSE* | Bias |
| TSS | S2 | $R865$ | 45 | 0.71 (e.) | 44 | 0.91 | 6.13 | 65.98 | 2.09 |
| | S3 | $R865$ | | 0.71 (e.) | | 0.91 | 6.03 | 64.87 | 2.01 |
| | S2 | $R783/R492$ | 45 | 0.90 (l.) | 41 | 0.93 | 4.57 | 49.69 | 0.27 |
| | S3 | $R779/R510$ | | 0.90 (l.) | | 0.94 | 4.35 | 47.22 | 0.38 |
| | S2 | NISSI ($R842$) | 45 | 0.90 (l.) | 44 | 0.84 | 6.74 | 72.71 | 0.15 |
| | S3 | NISSI ($R779$) | 42 | 0.92 (l.) | | 0.81 | 7.74 | 83.27 | 0.63 |
| TSS < 10 mg/L | S2 | $R665$ | 38 | 0.54 (l.) | 35 | 0.55 | 1.42 | 45.27 | 0.08 |
| | S3 | $R665$ | | 0.52 (l.) | | 0.55 | 1.71 | 54.42 | 0.86 |
| TSS < 20 mg/L | S2 | $R700$ | 40 | 0.84 (l.) | 37 | 0.85 | 1.79 | 42.89 | 0.39 |
| | S3 | $R700$ | | 0.84 (l.) | | 0.85 | 1.78 | 42.78 | 0.40 |
| | S2 | $R665/R560$ | 40 | 0.50 (p.) | 38 | 0.48 | 3.23 | 77.45 | 0.76 |
| | S3 | $R665/R560$ | | 0.48 (p.) | | 0.47 | 4.25 | 84.70 | 0.75 |
| TSS > 10 mg/L | S2 | $R665$ | 8 | 0.51 (l.) | 8 | 0.60 | 16.65 | 42.65 | 1.84 |
| | S3 | $R665$ | | 0.50 (l.) | | 0.59 | 16.71 | 43.12 | 2.00 |
| TSS > 20 mg/L | S2 | $R783/R492$ | 6 | 0.77 (l.) | 5 | 0.80 | 11.44 | 22.77 | 4.36 |
| | S3 | $R779/R510$ | | 0.78 (l.) | | 0.81 | 11.20 | 22.29 | 4.78 |
| | S2 | NISSI ($R842$) | 6 | 0.77 (l.) | 5 | 0.40 | 20.44 | 41.96 | 1.15 |
| | S3 | NISSI ($R779$) | | 0.98 (l.) | | 0.21 | 21.28 | 43.68 | 2.60 |

**Table 8.** Calibration and validation results for chlorophyll a (Chl_a). Only results with a calibration with $R^2 > 0.4$ are shown. Regressions: linear (l.); potential (p.); polynomial (pl.); exponential (e.). *RMSE* (mg/m$^3$); *RRMSE* (%); bias (mg/m$^3$).

| | Sensor | Bands Relation | Calibration | | Validation | | | | |
|---|---|---|---|---|---|---|---|---|---|
| | | | **n** | $R^2$ | **n** | $R^2$ | *RMSE* | *RRMSE* | **Bias** |
| Chl_a < 5 mg/m$^3$ | S2 | $\log_{10}$ [max. ($R443$; $R492$)/$R560$] | | 0.68 (l.) | | 0.55 | 0.94 | 43.18 | 0.09 |
| | S3 | $\log_{10}$ [max. ($R443$; $R490$)/$R560$] | 52 * | 0.69 (l.) | 53 | 0.57 | 0.92 | 42.43 | 0.08 |
| | S3 | $\log_{10}$ [max. ($R443$; $R490$; $R510$)/$R560$] | | 0.69 (l.) | | 0.57 | 0.93 | 42.82 | 0.13 |
| Chl_a > 5 mg/m$^3$ | S2 | $\left(\frac{1}{R665} - \frac{1}{R705}\right) \times R740$ | 73 | 0.92 (pl.) | 71 | 0.85 | 41.76 | 51.76 | 4.77 |
| | S3 | $\left(\frac{1}{R665} - \frac{1}{R709}\right) \times R754$ | | 0.91 (pl.) | | 0.84 | 79.35 | 98.36 | 29.87 |
| | S2 | $\frac{R705-R665}{705-665}$ | 73 * | 0.82 (pl.) | 70 | 0.73 | 56.04 | 73.47 | 1.15 |
| | S3 | $\frac{R709-R665}{709-665}$ | 73 | 0.83 (pl.) | | 0.82 | 54.97 | 72.07 | 14.44 |
| | S2 | $R740/R560$ | 73 | 0.91 (pl.) | 71 | 0.90 | 31.67 | 39.26 | 10.24 |
| | S3 | $R754/R560$ | | 0.91 (pl.) | | 0.89 | 43.67 | 54.17 | 10.72 |
| | S2 | $R705/R665$ | 72 | 0.93 (p.) | 71 | 0.91 | 35.16 | 41.07 | 2.99 |
| | S3 | $R709/R665$ | | 0.93 (p.) | | 0.91 | 37.10 | 43.95 | 3.40 |
| | S2 | $R665/R705$ | 73 * | 0.90 (e.) | 73 | 0.87 | 81.13 | 97.44 | 20.05 |
| | S3 | $R665/R709$ | | 0.89 (e.) | 72 | 0.87 | 60.13 | 71.50 | 15.82 |
| | S2 | $R705 - \frac{R665+R740}{2}$ | 73 * | 0.70 (p.) | 73 | 0.28 | 125.01 | 134.55 | 36.16 |
| | S3 | $R709 - \frac{R665+R754}{2}$ | | 0.72 (pl.) | 71 | 0.55 | 81.48 | 95.54 | 15.89 |
| | S2 | $\log_{10}$ [max. ($R443$; $R492$)/$R560$] | 73 * | 0.47 (pl.) | 73 | 0.21 | 128.01 | 137.87 | 36.99 |
| | S3 | $\log_{10}$ [max. ($R443$; $R490$)/$R560$] | | 0.50 (pl.) | | 0.23 | 124.86 | 134.38 | 33.83 |
| | S2 | $R705/R560$ | 72 * | 0.90 (pl.) | 71 | 0.89 | 39.74 | 48.00 | 2.22 |
| | S3 | $R709/R560$ | | 0.90 (pl.) | 72 | 0.90 | 37.04 | 43.88 | 0.49 |

* Using log10 of the data.

**Table 9.** Calibration and validation results for phycocyanin (PC). Only results with a calibration with $R^2 > 0.4$ are shown. Regressions: linear (l.); potential (p.). *RMSE* (mg/m$^3$); *RRMSE* (%); bias (mg/m$^3$).

| Sensor | Bands Relation | Calibration | | Validation | | | | |
|---|---|---|---|---|---|---|---|---|
| | | **n** | $R^2$ | **n** | $R^2$ | *RMSE* | *RRMSE* | **Bias** |
| S2 | $R705/R665$ | 70 | 0.78 (p.) | 64 | 0.79 | 43.67 | 54.92 | 14.64 |
| S3 | $R709/R679$ | 68 | 0.80 (p.) | 65 | 0.93 | 42.59 | 54.41 | 4.62 |
| S3 | $R709/R620$ | 67 | 0.77 (p.) | 63 | 0.82 | 34.57 | 46.75 | 7.03 |
| S2 | $R740/R665$ | 69 | 0.67 (p.) | 64 | 0.35 | 58.36 | 79.93 | 27.09 |
| S3 | $R754/R665$ | | 0.66 (p.) | 65 | 0.62 | 60.62 | 77.23 | 25.62 |
| S2 | $\left(\frac{1}{R665} - \frac{0.4}{R560} - \frac{0.6}{R705}\right) \times R740$ | 68 | 0.94 (l.) | 55 | 0.92 | 92.60 | 62.52 | 31.75 |
| S3 | $\left(\frac{1}{R620} - \frac{0.4}{R560} - \frac{0.6}{R709}\right) \times R754$ | 70 | 0.95 (l.) | 66 | 0.94 | 60.16 | 48.33 | 16.51 |
| S2 | Simis et al. [24] | 70 | 0.91 (l.) | 69 | 0.91 | 85.10 | 71.35 | 36.33 |
| S3 | Simis et al. [24] | 70 | 0.96 (l.) | 69 | 0.96 | 39.98 | 33.52 | 2.65 |

### 3.2.1. SDD

In the calibration process, the three linear functions obtained with field values and bands ratios showed a positive slope for SDD, as larger band ratio values correspond with clearer waters. All the tested band combinations had a similar coefficient of determination, with a maximum difference of 0.05 (Table 5). The $R490/R700$ ratio had the best coefficients of determination for S2 ($R^2 = 0.65$) and S3 ($R^2 = 0.64$), while the lowest coefficient of determination with a $R^2$ of 0.60 for both sensors was reached by the $R490/R560$ ratio.

Validation results showed higher differences between studied ratios. The best coefficients were found for the $R560/R700$ ratio, with a value of 0.77 for both sensors. This ratio

also registered the best *RMSE*, 1.03 m for S2 and 1.02 m for S3, and the best *RRMSE*, with percentage errors lower than 45%. The bias determined the lower values for the $R490/R560$ ratio, 0.10 m for S2 and 0.13 m for S3.

The *RMSE* calculated from the whole data set was always higher than 1 m. Since that error is an unacceptable error for SDD values below 1 m, the *RMSE* for the values of SDD lower than 1 m (n = 47) was calculated. The best results were obtained with the $R560/R700$ ratio, with an *RMSE* of 0.49 m. For the $R490/R700$ ratio, the *RMSE* was 0.67 m, and for the $R490/R560$ ratio, the *RMSE* was 1.09 m.

These results indicated that the best ratio for the retrieval of SDD was $R560/R700$, and, following the second step of calibration and validation process, the algorithm for the $R560/R700$ ratio was recalculated using the whole database, obtaining the results shown in Figure 4.

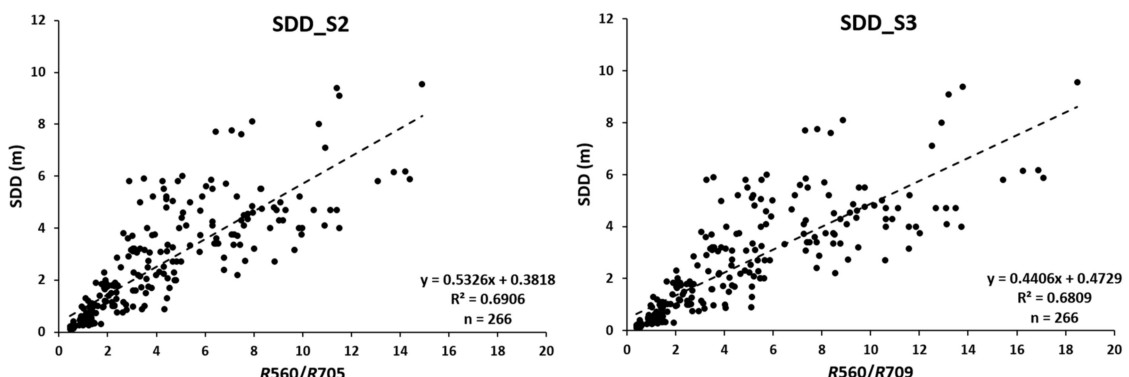

**Figure 4.** SDD algorithms recalculated using all database.

### 3.2.2. CDOM

According to the bibliography cited in the introduction section, both blue–red and green–red ratios were applied to S2 and S3 spectral bands. Different types of functions in the calibration regressions were obtained and the summary of results is given in Table 6.

The results were very similar for five different ratios. While, in calibration regression models, the best ratios were two green–red ratios ($R560/R665$ and $R560/R700$) with a $R^2$ of 0.51 for S2 and a $R^2$ of 0.50 for S3, for error statistics, the best results were given in a blue–red ratio.

The algorithm with better results in the validation process was $R665/R490$, obtaining the lowest values for *RMSE*, *RRMSE* and bias. Using S2 convoluted bands, the *RMSE* was 0.88 µg/L QSE, the only one below 0.9 µg/L QSE, corresponding to a *RRMSE* of 47.94%. For S3, *RMSE* was a little higher, 0.9 µg/L QSE, with a *RRMSE* of 48.90%, the only one below 50% for S3 results. The bias was the same for two sensors, 0.03 µg/L QSE. The coefficients of determination of 0.53 for S2 and 0.51 for S3 were the most approximated to the higher value for both, 0.55, obtained for the $R560/R700$ ratio.

These results indicated the $R665/R490$ ratio as the more suitable ratio for retrieval CDOM in our dataset, and, following the second step of calibration and validation process, the algorithm for the $R665/R490$ ratio was recalculated using the whole database, obtaining the results shown in Figure 5.

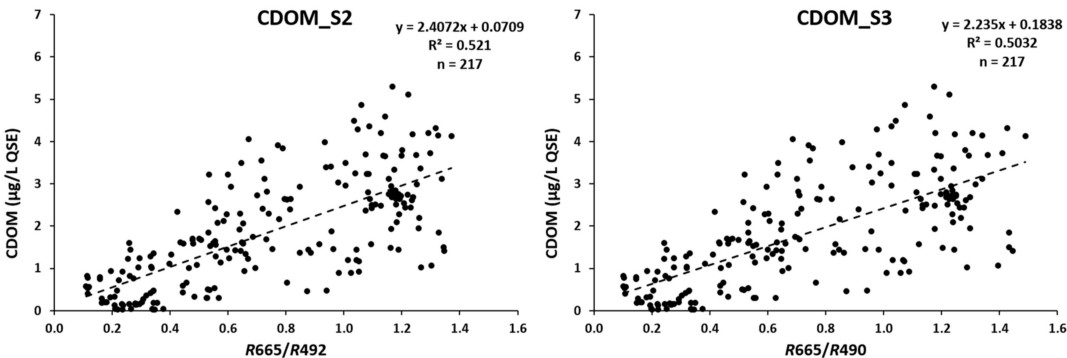

**Figure 5.** CDOM algorithms recalculated using all database.

3.2.3. TSS

Initially, for the TSS retrieval algorithms, the whole calibration data set was used, but, finally, the data set according to the TSS concentration were separated. Table 7 presents the best results, where all calibration regression models present linear functions except one exponential and one potential.

With regards to the calibration process, when the whole calibration data set was used, the best results were two linear functions with coefficients of determination of 0.9 or higher, the $R783/R492$ ratio for S2 and $R779/R510$ for S3, and the NIR bands Suspended Sediment Index (NISSI), using $R842$ for S2 and $R779$ for S3. The validation process determined the better results for the $R783/R492$ ratio, with a *RMSE* of 4.57 and 4.35 mg/L for S2 and S3, respectively, and a *RRMSE* of 49.69% and 47.22%.

The results using only part of the data range, concentrations under 10 mg/L or under 20 mg/L, have demonstrated that, with the $R700$ band only, it is possible to reduce the *RRMSE* down to 42% with a *RMSE* of 1.79 and 1.78 mg/L and a bias of 0.39 mg/L and 0.40 mg/L for S2 and S3, respectively.

The best retrieval algorithm using only the high values of our range data was the same as using all data, the $R783/R492$ ratio for S2 and $R779/R510$ for S3. However, the high values of concentration caused an increase of two error statistics in contrast with the results using all data. The higher values led to the *RMSE* increasing to 11 mg/L and the bias achieving values of 4 mg/L, although the *RRMSE* descended until 22%.

With the aim of reducing errors, these results lead us to choose two algorithms, the simple band $R700$ for low TSS concentrations, under 20 mg/L, and the $R783/R492$ ratio for S2 and $R779/R510$ for S3 for high TSS concentrations, above 20 mg/L. Following the second step of the calibration and validation process, the algorithms using the whole database were recalculated, obtaining the results shown in Figures 6 and 7. A threshold has been determined to facilitate the algorithm application for concentrations above 20 mg/L. If the $R783/R492$ ratio for S2 and the $R779/R510$ ratio for S3 surpassed the value of 0.8, the algorithm for concentrations above 20 mg/L were applied.

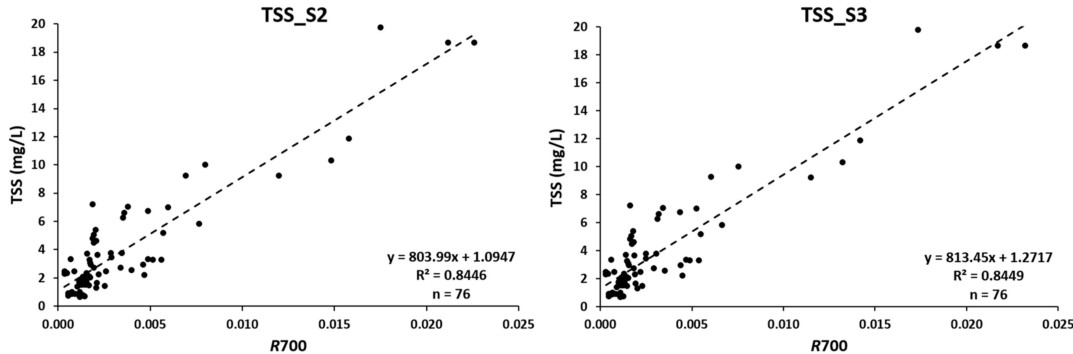

**Figure 6.** TSS algorithms recalculated using all data under 20 mg/L.

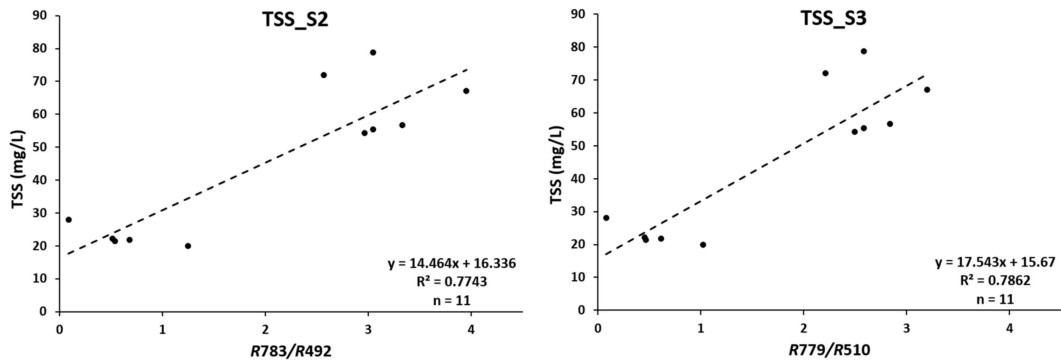

**Figure 7.** TSS algorithms recalculated using all data upper 20 mg/L.

### 3.2.4. Chl_a

The difficulties to obtain an algorithm with accurate error for low values using the entire calibration data range led us to separate data according to the Chl_a concentration. As shown in Table 8, the values under 5 mg/m$^3$ were separated from the rest. Calibration regression models were linear functions for low values and polynomial for most regression models of high values.

Only one band combination achieved good results when using data with concentrations below 5 mg/m$^3$, the correlation between the log10 ratio using maximum value of $R443$ or $R490$ and $R560$ and the log10 of Chl_a concentrations. Coefficient of determination for the calibration regression model near to 0.7 was obtained, while, for the validation regression model, the coefficient descended until 0.55 for S2 and 0.57 for S3. On the other hand, the $RMSE$ were over 0.9 mg/m$^3$, corresponding to a $RRMSE$ of 43.18% for S2 and 42.43% for S3. The bias was less than 0.1 mg/m$^3$.

Using data above 5 mg/m$^3$, four calibration regression models reached a coefficient of determination of 0.9 or higher: the triband model, other bands operation and two ratios. However, the validation regression models with a coefficient of determination of 0.9 was obtained only when using ratios. The best results were obtained with the $R700/R665$ ratio reaching a $RMSE$ of 35.16 mg/m$^3$, equivalent to a $RRMSE$ of 41.07% and a bias of 2.99 mg/m$^3$ for S2, and a $RMSE$ of 37.10 mg/m$^3$, equivalent to a $RRMSE$ of 43.95% and a bias of 3.40 mg/m$^3$ for S3.

These results indicated the ratio using the maximum value of $R443$ or $R490$ and $R560$ as the more suitable ratio for retrieval of Chl_a concentrations below 5 mg/m$^3$. Meanwhile, for concentrations above 5 mg/m$^3$, these results led us to choose the $R700/R665$ ratio for S2 and S3. Following the second step of the calibration and validation process, the algorithms were recalculated using the whole database, obtaining the results shown in Figures 8 and 9. A threshold was determined to facilitate the algorithm application for concentrations above 5 mg/m$^3$. If the $R700/R665$ ratio for S2 and S3 surpassed the value of 0.8, the algorithm for concentrations above 5 mg/m$^3$ was applied.

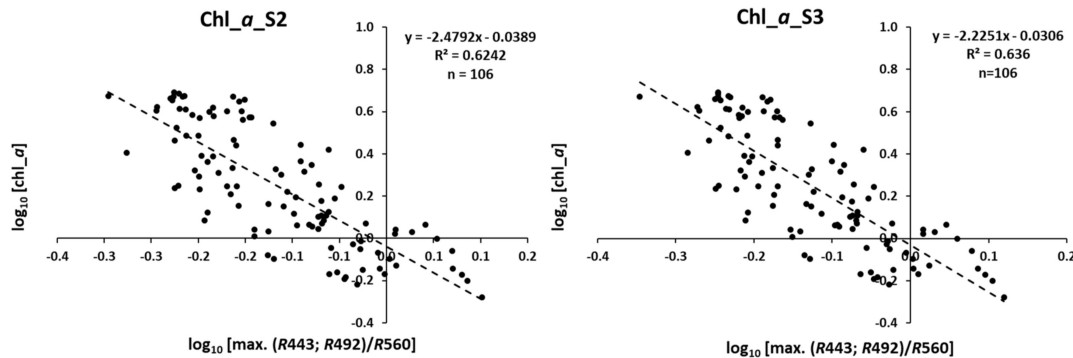

**Figure 8.** Chl_a algorithms recalculated using all data under 5 mg/m$^3$.

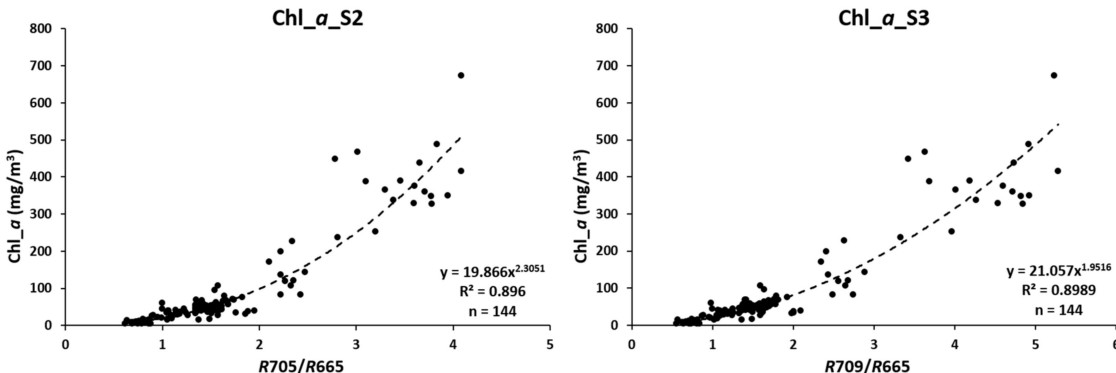

**Figure 9.** Chl_a algorithms recalculated using all data upper 5 mg/m$^3$.

### 3.2.5. PC

For the phycocyanin retrieval algorithm, only two of the five calibration regression models reached a coefficient of determination higher than 0.9, the four-band model and the semi-analytical approach (Table 9). These were linear functions, compared to the rest that were potential.

With regards to S2, the algorithm with better results in the validation process was $R705/R665$, with a *RMSE* of 43.67 mg/m$^3$, a *RRMSE* of 54.92% and a bias of 14.64 mg/m$^3$. Note that this was not the ratio with the highest coefficient of determination. While for S3, with the band 620 that corresponded to the maximum PC absorption, the best results were reached with the semi-analytical approach with a coefficient of determination of 0.96, the *RMSE* was 39.98 mg/m$^3$, corresponding to a *RRMSE* of 33.52 and a bias of 2.65 mg/m$^3$.

Following the second step of calibration and validation process, the algorithm of the $R705/R665$ ratio was recalculated for S2 using the whole database, obtaining the results shown in Figure 10. In the semi-analytical approach, all the data were used to recalculate the specific absorption coefficient of PC at waveband 620, $a_{PC}^*(620)$, determined as the absorption coefficient of PC at waveband 620, $a_{PC}(620)$, divided by the measured PC concentration. While Simis et al. [24] used 0.007 m$^2$/mg, in the present study, a value of 0.0034 m$^2$/mg was obtained. Estimated and observed PC concentrations had a strong correlation, with a coefficient of determination of 0.95, as shown Figure 10.

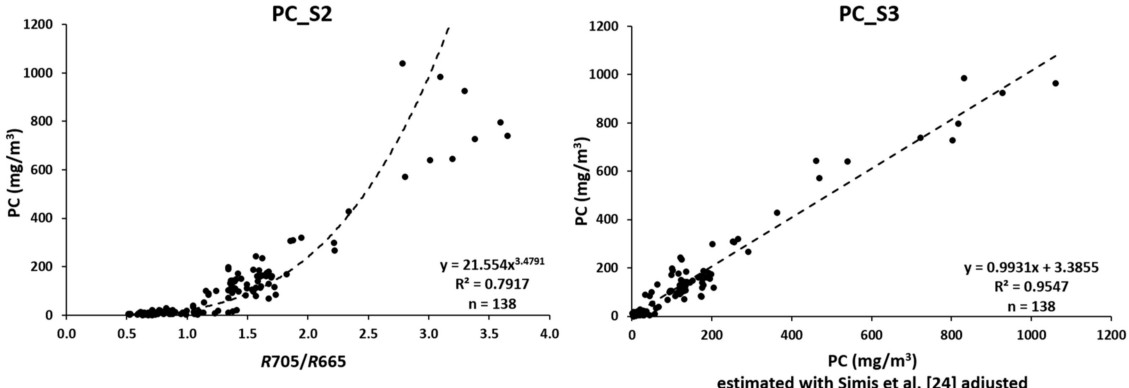

**Figure 10.** PC algorithms recalculated using all database.

Summary of retrieval algorithms of each studied variable using all data base is given in Table 10. It is possible to see the used data range, number of used data, obtained algorithm, the coefficient of determination and the *RMSE*.

**Table 10.** Summary of retrieval algorithms.

| Parameter | Range | $n$ | SENTINEL 2 | | | SENTINEL 3 | | |
|---|---|---|---|---|---|---|---|---|
| | | | Equation | $R^2$ | RMSE | Equation | $R^2$ | RMSE |
| SDD (m) | 0.1–9.55 | 266 | $0.5326 \times (R560/R705) + 0.3818$ | 0.69 | 1.14 | $0.4406 \times (R560/R709) + 0.4729$ | 0.68 | 1.16 |
| CDOM (μg/L QSE) | 0.03–5.30 | 217 | $2.4072 \times (R665/R492) + 0.0709$ | 0.52 | 0.88 | $2.235 \times (R665/R490) + 0.1838$ | 0.50 | 0.90 |
| TSS (mg/L) | 0.67–19.76 | 76 | $803.99 \times R700 + 1.0947$ | 0.85 | 1.55 | $813.45 \times R700 + 1.2717$ | 0.85 | 1.55 |
| | 20.00–78.82 | 11 | $14.464 \times (R783/R492) + 16.336$ | 0.77 | 10.35 | $17.543 \times (R779/R510) + 15.67$ | 0.79 | 10.07 |
| Chl_a (mg/m$^3$) | 0.53–4.92 | 106 | $\exp.10(-2.4792 \times (\log_{10}[\max.(R443;R492)/R560]) - 0.0389)$ | 0.62 | 0.91 | $\exp.10(-2.2251 \times (\log_{10}[\max.(R443;R490)/R560] - 0.0306)$ | 0.64 | 0.90 |
| | 5.16–674.70 | 144 | $19.866 \times (R705/R665)^{2.3051}$ | 0.90 | 35.68 | $21.057 \times (R709/R665)^{1.9516}$ | 0.90 | 37.29 |
| PC (mg/m$^3$) | 0.13–1040 | 138 | $21.554 \times (R705/R665)^{3.4791}$ | 0.79 | 44.48 | $294 \times \left( \left( \left\{ \left[ \frac{R709}{R620} \right] \times [0.727 + b_b] \right\} - b_b - 0.281 \right) - \left[ \varepsilon \times a_{ph}(665) \right] \right)$ $b_b(779) = \frac{1.61 \times R779}{\{0.082 - [0.6 \times R779]\}}$ $a_{ph}(665) = 1.47 \times \left( \left\{ \left[ \frac{R709}{R665} \right] \times [0.727 + b_b] \right\} - 0.401 - b_b \right)$ | 0.96 | 41.47 |

## 4. Discussion

The possibility of bringing together two large databases, obtained from two projects carried out by this manuscript's co-authors such as CEDEX project and ESAQS project, allowed us to work with a large amount of high-quality data. Furthermore, its high heterogeneity made it possible to work within a wide range in each of the variables, so that the algorithms could be used in equally heterogeneous water bodies. However, the data sets are not distributed evenly throughout the range; low values are more numerous for all variables which gives them greater weight in retrieval algorithms.

This study presents a very extensive database of inland water quality data, which coincides with field spectroradiometry, allowing us to test all possible retrieval algorithms for the OLCI and MSI sensors. The database covers a wide gradient of limnological variables and is representative of the variability of climatic and limnological conditions of the Mediterranean Basin, with a greater representation of semi-arid environment reservoirs (with high residence times, which determine long stratification periods and a tendency for summer Cyanobacterial blooms). This amount of data encouraged us to research general empirical algorithms for all variables, although, for TSS and Chl_a, it was only possible by separating the data set into two concentration ranges. However, the extensive revision of models carried out allowed us to determine which ones are the most accurate for the remote sensing studies of these variables using S2 and S3, taking into account the following reflections.

The spectra of the studied water bodies are mostly influenced by TSS and phytoplankton pigments. These are the OACs mainly affecting water transparency and, thus, determining the bands better correlated with this variable. In fact, as the strong correlation of TSS and Chl-a indicates, the main driver of the overall optical properties in this data set is the phytoplankton. In contrast, CDOM has less influence in water transparency and in the spectral features of the data set.

SDD is affected by all optically active substances combined in the water (Chl_a, CDOM and other substances); therefore, it may be difficult to find conclusive reasons for using some particular wavelengths for retrieval SDD rather than others [56]. Despite the difficulties to obtain the SDD retrieval algorithm, all data could be used together to obtain a particular band ratio for two sensors, the $R560/R700$ ratio, widely used in other sensors like MERIS. The coefficient of determination obtained with all the data was 0.7 (Figure 4), a value very close to those obtained in the previous studies considered (Table 2) that use satellite image data. We also wanted to compare calibration correlation slopes and offsets values with consulted manuscripts that apply the same combination bands and fitting function. Our slopes are 0.53 (S2) and 0.44 (S3), while our offsets are 0.38 (S2) and 0.47 (S3). These are better results than in a study carried out in a hypertrophy lagoon with a slope of 0.22 and offset of 0.084 [28] but worse than a study in clear waters with a slope of 0.99 and offset of 0.34 [13], both using satellite image data.

It was also the most accurate ratio in the works of Sòria-Perpinyà et al. [28] and Soomets et al. [27] using satellite image data. These were based on previous works with MERIS sensor, in the first case, with the premise that SDD and Chl_a are inversely related, and some algorithms to obtain Chl_a have compared the ratio of the reflectance at the peak around 700 nm to the reflectance at the peak around 560 nm, which is related to a minimum in the combined absorption by phytoplankton pigments, particles and CDOM [57], and in the second case, in the light attenuation coefficient for inland waters of Alikas et al. [58].

For CDOM, it was also possible use all of the data together, obtaining the best correlation for both sensors with the $R665/R490$ ratio. The coefficient of determination was low, 0.5 (Figure 5), which could be due to the lesser influence of CDOM on the transparency for our database. Although the errors obtained are among 47 and 49%, an assumable error since the values referred to are low, and therefore, the error is small, only 0.9 µg/L QSE. This red–blue ratio is consistent with the results of Soomets et al. [27], using satellite image data, and Ruescas et al. [40], using simulated data. To compare our calibration correlation slopes and offsets values with consulted manuscripts that apply same combination bands

and fitting function, our slopes are 2.41 (S2) and 2.24 (S3) and our offsets are 0.07 (S2) and 0.18 (S3). These values are inside slopes and offsets ranges of a study carried out using satellite image data and a classification of five different optical water types, with slopes range of 0.26–21.3 and offsets range of 1.56–9.67 [27].

Therefore, although is true that in situ data does not need any correction and those of Ruescas et al. [40] are simulated, it is also true that both Soomets et al. [27] and his work of reference [59] use atmospherically corrected data, so it is possible that, despite atmospheric correction errors in the blue region, we can use it for CDOM determination. In contrast, all papers consulted are in agreement with the suggestion of Zhu et al. [60] about the fact that empirical algorithms in complex inland waters can be significantly improved by selecting at least one band with a relatively longer wavelength (>600 nm). It reduces the possible effects of particulate matter, and, in inland waters, chlorophyll and nonalgal particles are usually in high concentrations and present high backscattering within the longer wavelengths.

Two models had to be adjusted for TSS, one for low and one for high concentrations, in order to reduce algorithm errors. Another possibility is that different TSS compositions present different slopes, resulting from different inherent optical properties, and, thus, due to different characteristics of suspended material [61]. The absence of loss on ignition (method to estimate the organic content) data from CEDEX project did not let us to improve respective influences of the organic and inorganic suspended matter on the relationship. However, of the 11 data used in the correlation, only 1 did not correspond to a hypereutrophic lake.

The algorithm for low values, under 20 mg/L, was obtained by a simple band $R700$ for the two sensors, with a coefficient of determination of 0.85 (Figure 6). The accuracy of the algorithms is also ratified by the validation regression equations, because the slope value is very close to 1 and the intercept value close to 0. The band $R700$ used matches for different water types in the Soomets et al. [27] manuscript using satellite images, although it uses few values higher than 30 mg/L, one in MSI sensor and three in OLCI sensor, two of which do not fit in validation regression models. If we compare our calibration correlation slopes of 803.99 (S2) and 813.45 (S3) and offsets of 1.09 (S2) and 1.27 (S3), we can observe that they are similar to theirs, with a slopes range of 46.19–531 and an offsets range of 0.15–16.26 [27]. Furthermore, if we consult the works revised by them to test this band, none of them use high TSS values (<32 mg/L). Only the Kallio et al. [62] manuscript uses a range of TSS similar to ours, with concentrations between 0.7 and 32 mg/L. We can see that their results were very similar to ours, using airborne imaging spectrometry with 92 samples they obtained a $R^2$ of 0.82, a *RMSE* of 3.15 mg/L and a *RRMSE* of 33%.

Another algorithm was determined, using data higher than 20 mg/L, to have sufficient data for calibration and validation processes. The best correlation was defined with the $R783/R492$ ratio for S2 and $R779/R510$ for S3, with a coefficient of determination of 0.8 (Figure 7). Despite having fewer samples for calibration and validation processes, our results are very similar to Yuan et al. [52], a manuscript tested in this research, and, for OLCI sensor, obtained a validation regression model with a $R^2$ of 0.9 and a *RMSE* of 19.29 mg/L using satellite images and data range of 33.88-695.24 mg/L.

The other variable with two algorithms was Chl_a. The threshold value was 5 mg/m$^3$, because waters with high Chl_a concentrations (above 3–5 mg/m$^3$ [63]) produce discernible spectral features in the red and NIR regions of the reflectance spectrum [12].

From the 14 band combinations tested for Chl_a, for values under 5 mg/m$^3$, only two reached a coefficient of determination higher than 0.5. The algorithm with best results was obtained relating log10 of the ratio using the highest value among $R443$ and $R492$ bands and $R560$ band with log10 Chl_a concentrations, reaching a coefficient of determination for S2 of 0.62 and 0.64 for S3 (Figure 8). Our validation results with a *RMSE* of 0.94 mg/m$^3$ (S2) and 0.92 mg/m$^3$ (S3) are in concordance with Pereira et al. [13] using satellite images, with a mean absolute error of 0.89 mg/m$^3$, using values under 10 mg/m$^3$, despite the fact that 35 of 43 used values were under 5 mg/m$^3$. The used bands are in concordance

with the bands of highest value for under 5 mg/m$^3$ concentrations in the study developed by O'Reilly and Werdell [55] using the highest value of $R443$, $R492$ or $R510$ bands like a numerator, obtaining a $R^2$ of 0.86 in the validation of a Chl_a concentration wide range (0.012 to 77.9 mg/m$^3$). However, their study was carried out in ocean and coastal waters, while we studied inland waters, which are more optically complex.

For values above 5 mg/m$^3$ the best results obtained were for the $R700/R665$ ratio, with a coefficient of determination of 0.9 (Figure 9). It was also the best algorithm in four of the ten checked studies: Moses et al. [12], Xu et al. [14], Chen et al. [15] and Lins et al. [16]. These studies use a single algorithm for a Chl_a concentration range from 1 to 120 mg/m$^3$, while our range is from 5 to 700 mg/m$^3$. Is possible that, for these reasons, our $RMSE$ is higher, 35.16 mg/m$^3$ (S2) and 37.10 mg/m$^3$ (S3), and in the studies mentioned, it only reaches 10 mg/m$^3$. If we compare our $RRMSE$ of 41% for S2 and 44% for S3 using 71 samples, in regards to works where they calculated it, our results are similar to the $RRMSE$ of 48% obtained by Chen et al. [15], using 41 samples applying a potential fit, and much bigger than that obtained by Moses et al. [12] of only 8%, using only 15 samples applying a lineal fit. We applied a potential fit, and the result was that our calibration correlation slopes of 2.31 for S2 and 1.95 for S3 and offsets of 19.87 for S2 and 21.06 for S3 are very similar to those obtained by Chen et al. [15], with a slope of 3.12 and an offset of 25.99.

PC is the only variable with two clearly different algorithms for S2 and S3. This is mainly due to the $R620$ band, as mentioned in the results, and also due to the fact that the narrow bands of OLCI sensor allow the application of a semi-analytical model.

The best algorithm for S2 was obtained with the $R700/R665$ ratio, with a coefficient of determination of 0.8 (Figure 10). This is the same band ratio as the one obtained to estimate Chl_a concentrations above 5 mg/m$^3$, due to the strong correlation between these two variables with a Spearman correlation coefficient, 0.848. We used the same combination of bands but not the same equation. The position and bandwidths of S2 are not optimal to detect most of the specific features (peaks, shoulders, troughs) caused by the water optically active constituents (OAC) in the water-leaving reflectance. This is the case of Cyanobacteria, whose characteristic trough at 620 nm, due to phycocyanin absorption, is only detectable in S3–OLCI bands. In S3, the estimation of total phytoplankton and Cyanobacteria biomasses (through Chl_a and PC, respectively) are, therefore, uncorrelated, since they use different band sets, while, in S2, there will be always a correlation in the satellite products, which does not necessarily correspond to the actual correlation of the variables in the water bodies. On other words, an increase in Chl_a can be caused by an increase of other phytoplankton types, such as green algae, different from PC-rich Cyanobacteria.

This issue reinforces one of the main purposes of this work, which was to explore the complementarities of S2 and S3 for inland water quality monitoring. S2 allowed the study of surface dynamics of a large number of water bodies, thanks to its spatial resolution, but with often less accurate or limited algorithms than S3, which provides higher spectral and temporal resolutions.

The obtained ratio does not coincide exactly with any of the five works reviewed, although it could be considered the equivalent of S2 to the obtained by Beck et al. [22] for S3 with $R707/R679$ ratio using in situ reflectance, because S3-centered band at 679 nm (677–685 nm) was replaced for S2-centered band at 665 nm (649–680 nm). Unfortunately, the difference in the PC concentration units used did not allow us to compare our results with this study, so we compared them to two studies with a similar range of data. The four-band model developed for Liu et al. [23] using satellite images, obtain a $RMSE$ of 27.69 mg/m$^3$ and a mean absolute percentage error of 172.863% with a data range of 0.33–317.74 mg/m$^3$, while our algorithm percentage error calculated as $RRMSE$ is much lower, 54.92%, although it has a higher $RMSE$ of 43.67 mg/m$^3$. The opposite happens in the study of Sòria-Perpinyà et al. [21] using satellite images, with a $RMSE$ of 141 mg/m$^3$ but a $RRMSE$ of 40%, a lower error by having a greater amount of data with high values, since its average value is 405 mg/m$^3$ and ours is 106 mg/m$^3$.

The recalibrated Simis et al. [24] semi-analytical approach was the best model to estimate PC through the OLCI sensor, with a coefficient of determination of 0.9 (Figure 10). The difference between $a_{PC}^*(620)$ of 0.007 m$^2$/mg, used by Simis et al. [24], and that obtained with our results of 0.0034 m$^2$/mg could be explained by the higher PC concentrations and conditions of higher irradiance in our study area, in comparison to the data obtained in The Netherlands lakes.

The algorithms defined in this work could be very valuable tools for water quality monitoring, especially in Mediterranean climate areas were the climatic models forecast predicts a water deficit. As a consequence of a water deficit, reservoirs will be more sensitive to a eutrophication process, because increased evapotranspiration leads to higher nutrient concentrations in the remaining water [64].

In addition, particularly in our study area, our algorithms will facilitate the incorporation of applied remote sensing into the hydrographic basins monitoring programs of Iberian Peninsula. Monitoring programs in compliance with Water Framework Directive require a minimum frequency of data, which is difficult and costly to achieve with only field data. Remote sensing data would serve to perform more frequent monitoring by applying the models proposed in this work for the estimation of key variables to determinate inland waters ecological status.

Further validation to test the application of these algorithms must be addressed, considering the obstacles when trying to transfer the results to real satellite remote sensing data, such as atmospheric and sun specular correction. To have some indication on the potential problems and/or errors when applying the models to real remote sensing data, we did a limited validation in a well-known reservoir (Alarcón), in which, despite not having enough matchup data with satellite acquisitions, we had a good knowledge of the usual ranges of the variables and their spatial distribution (Appendix B). The values obtained were within the expected range in the Alarcón images for Chl_a (Figure A1) and TSS (Figure A2).

## 5. Conclusions

In this work, we proposed and validated retrieval methods for five key variables (Chl_a, PC, SDD, TSS and CDOM) from MSI and OLCI sensors using a large in situ data set representative of lakes and reservoirs of the Mediterranean basin. The data set included radiometric and water quality data obtained in the periods 2001–2007 and 2017–2018.

Our goal was to provide algorithms for S2 and S3 to exploit the combined use of both satellites for the continuous monitoring of the spatial and temporal variation of key variables determining inland waters' ecological status. The best algorithms were empirical and used similar bands for both sensors, with the exception of PC algorithm for S3. The narrower bands of OLCI facilitated finding specific features in the water-leaving radiance and allowed the application of a semi-analytical model using its specific band at 620 nm that corresponds to the maximum PC absorption. The S3 algorithms can serve as a further validation of the S2 algorithms when the spatial consistency of both products is studied. This would require study of a large number of small- and medium-sized water bodies that could not be studied with S3.

Regarding the calibration and validation strategy with different data sets but same ranges for the variables' values, the retrieval algorithms demonstrated to be consistent and suitable to estimate key variables of water quality. The recalibration of the algorithms using all the database enhanced the algorithms robustness.

A preliminary test with satellite image in a well-known reservoir showed results consistent with the expected ranges and spatial patterns of the variables. The selected models could serve in future works to map five water quality variables from remotely sensed data using S2 and S3, both separated and combined.

**Author Contributions:** Conceptualization, X.S.-P., A.R.-V. and J.D.; Data curation, P.U. and C.T.; Formal analysis, J.M.S; Funding acquisition, J.M.; Investigation, X.S.-P., P.U., M.P.-S., A.R.-V. and R.P.; Methodology, X.S.-P.; Project administration, J.D.; Resources, E.V., J.M.S. and J.M.; Software, P.U. and

C.T.; Supervision, J.M.; Validation, X.S.-P., M.P.-S. and R.P.; Visualization, P.U.; Writing—original draft, X.S.-P.; Writing—review & editing, E.V. and A.R.-V. All authors have read and agreed to the published version of the manuscript.

**Funding:** This research was partially funded by the European Union—ERDF and the Ministry of Science and Innovation and the State Research Agency of Spain under project RTI2018-098651-B-C51 (FLEXL3L4—Advanced Products L3 and L4 for the FLEX-S3 mission) and partially funded by the GENERALITAT VALENCIANA postdoc research grant APOSTD/2020/134, the project SEQUARMON (Sentinel quality reservoirs monitoring).

**Institutional Review Board Statement:** Not applicable.

**Informed Consent Statement:** Not applicable.

**Data Availability Statement:** The data presented in this study are available on request from the corresponding author.

**Acknowledgments:** The authors especially acknowledge the work done for obtaining the CEDEX database. Thanks to Manuel Toro, head of the limnology area in Centre for Hydrographic Studies (CEDEX, Spain) for kindly providing access to the data. Very special thanks to Ramón Peña, who was responsible for all campaigns and data processing leading to this valuable database, as well as the team of researchers, drivers, field and laboratory technicians that participated. Authors would also like to thank anonymous reviewers for their constructive comments and suggestions.

**Conflicts of Interest:** The authors declare no conflict of interest. The funders had no role in the design of the study; in the collection, analyses, or interpretation of data; in the writing of the manuscript or in the decision to publish the results.

## Appendix A

**Table A1.** Summary of sampled lakes and reservoirs. Symbols and abbreviations: max.: maximum; m.a.s.l.: meters above sea level; lat.: latitude; lon.: longitude; Res.: residence. [a] Visits spanning 2 days counted as 1.

| Name | Position | | Depth | Volume | Elevation | Res. Time | Climate | Visits [a] | Samples | # of Spectroradiometry Samples | | | # of Variables Samples | | | | |
|---|---|---|---|---|---|---|---|---|---|---|---|---|---|---|---|---|---|
| | Lat. | Lon. | m (max.) | ×10$^6$ m$^3$ | m.a.s.l. | years | | | | ASD-FR | HandHeld 2 | HR4000 | SDD | CDOM | TSS | Chl_a | PC |
| | d.dd | d.dd | | | | | | | | | | | m | µg/L QSE | mg/L | mg/m$^3$ | mg/m$^3$ |
| Aguilar | 42.80 | −4.32 | 48 | 247 | 942 | 0.78 | Cfb | 2 | 4 | 4 | | | 4 | 4 | 0 | 4 | 4 |
| Alarcón | 39.60 | −2.17 | 71 | 1118 | 806 | 2.15 | Csa | 2 | 4 | 4 | | | 4 | 4 | 0 | 4 | 4 |
| Albufera | 39.34 | −0.35 | 2 | 360 | 1 | 10.0 | Csa | 7 | 36 | 29 | 7 | | 36 | 18 | 6 | 28 | 19 |
| Alcántara | 39.75 | −6.75 | 135 | 3200 | 218 | 0.43 | Csa | 2 | 8 | 8 | | | 8 | 2 | 4 | 7 | 2 |
| Alcorlo | 41.02 | −3.02 | 62 | 180 | 920 | 2.22 | Csa | 1 | 1 | 1 | | | 1 | 1 | 0 | 1 | 1 |
| Almendra | 41.22 | −6.28 | 202 | 2413 | 730 | 1.57 | Csb | 3 | 13 | 13 | | | 13 | 3 | 6 | 9 | 3 |
| Bellús | 38.93 | −0.47 | 34 | 69 | 144 | 0.24 | Csa | 3 | 7 | | 6 | 1 | 7 | 6 | 7 | 7 | 5 |
| Benaixeve | 39.73 | −1.09 | 90 | 221 | 450 | 0.63 | Csb | 4 | 13 | | 13 | | 13 | 13 | 13 | 13 | 5 |
| Beniarrés | 38.80 | −0.35 | 53 | 27 | 318 | 0.35 | Csa | 3 | 4 | 1 | 3 | | 4 | 4 | 3 | 4 | 2 |
| Bornos | 36.80 | −5.73 | 52 | 215 | 104 | 0.72 | Csa | 1 | 2 | 2 | | | 2 | 2 | 0 | 2 | 2 |
| Brovales | 38.35 | −6.68 | 25 | 7 | 303 | n/a | Csa | 1 | 1 | 1 | | | 1 | 1 | 0 | 1 | 1 |
| Buendía | 40.40 | −2.77 | 79 | 1458 | 712 | 2.73 | Csa | 1 | 1 | 1 | | | 1 | 1 | 0 | 1 | 1 |
| Burguillo | 40.42 | −4.60 | 91 | 198 | 729 | 0.47 | Csa | 2 | 4 | 4 | | | 4 | 4 | 0 | 4 | 4 |
| Canelles | 42.03 | 0.65 | 150 | 201 | 506 | 1.22 | Cfb | 1 | 4 | 4 | | | 4 | 4 | 0 | 4 | 0 |
| Cernadilla | 42.02 | −6.47 | 69 | 233 | 889 | 0.47 | Csb | 1 | 1 | 1 | | | 1 | 1 | 0 | 1 | 1 |
| Cijara | 39.37 | −5.00 | 80 | 1470 | 428 | 1.68 | Csa | 1 | 2 | 2 | | | 2 | 2 | 0 | 2 | 2 |
| Contreras | 39.62 | −1.53 | 129 | 852 | 669 | 1.48 | Csa | 5 | 17 | 2 | 8 | 7 | 15 | 12 | 13 | 15 | 3 |
| Cortes | 39.23 | −0.92 | 112 | 118 | 326 | 0.08 | Csa | 1 | 1 | 1 | | | 1 | 1 | 0 | 1 | 1 |
| Cuerda del Pozo | 41.85 | −2.75 | 40 | 200 | 1,078 | 1.96 | Cfb | 2 | 7 | 7 | | | 7 | 0 | 6 | 6 | 0 |
| Ebro | 42.97 | −4.07 | 34 | 540 | 838 | 1.55 | Cfb | 2 | 3 | 3 | | | 3 | 3 | 0 | 2 | 2 |
| El Atazar | 40.90 | −3.53 | 141 | 426 | 873 | 1.19 | Csa | 3 | 4 | 4 | | | 4 | 3 | 0 | 4 | 4 |
| Entrepeñas | 40.50 | −2.72 | 85 | 874 | 718 | 1.04 | Csa | 1 | 1 | 1 | | | 1 | 1 | 0 | 1 | 1 |
| Giribaile | 38.08 | −3.48 | 84 | 475 | 346 | 1.28 | Csa | 1 | 1 | 1 | | | 1 | 1 | 0 | 1 | 1 |
| Guadalcacín | 36.65 | −5.75 | 44 | 800 | 102 | 2.40 | Csa | 1 | 2 | 2 | | | 2 | 2 | 0 | 2 | 2 |
| Guadalén | 38.17 | −3.47 | 55 | 173 | 350 | 1.25 | Csa | 1 | 1 | 1 | | | 1 | 1 | 0 | 1 | 1 |
| Guadalteba | 36.95 | −4.83 | 84 | 173 | 362 | 1.44 | Csa | 1 | 2 | 2 | | | 2 | 2 | 0 | 2 | 2 |
| Iznájar | 37.25 | −4.30 | 120 | 980 | 421 | 1.63 | Csa | 3 | 14 | 14 | | | 14 | 4 | 6 | 9 | 3 |
| Jándula | 38.25 | −3.92 | 88 | 322 | 360 | 1.29 | Csa | 1 | 2 | 2 | | | 2 | 2 | 0 | 2 | 2 |
| La Serena | 38.88 | −5.17 | 89 | 2828 | 355 | 3.59 | Csa | 2 | 4 | 4 | | | 4 | 4 | 0 | 4 | 4 |
| Maria Cristina | 40.02 | −0.16 | 38 | 18 | 100 | 5.96 | Csa | 2 | 5 | | 5 | | 5 | 4 | 4 | 4 | 0 |
| Navalcán | 40.03 | −5.10 | 26 | 34 | 370 | 0.46 | Bsk | 1 | 3 | 3 | | | 3 | 3 | 0 | 3 | 3 |
| Negratín | 37.55 | −2.93 | 75 | 496 | 638 | 1.87 | Csb | 1 | 2 | 2 | | | 2 | 2 | 0 | 2 | 2 |
| Pinilla | 40.93 | −3.55 | 33 | 38 | 1089 | 0.27 | Csb | 3 | 4 | 4 | | | 4 | 4 | 0 | 4 | 3 |
| Regajo | 39.89 | −0.52 | 23 | 6 | 407 | 0.14 | Csa | 2 | 7 | | 4 | 3 | 7 | 7 | 7 | 7 | 4 |
| Rialb | 41.97 | 1.23 | 99 | 402 | 430 | 0.36 | Cfa | 1 | 4 | 4 | | | 4 | 4 | 0 | 4 | 0 |
| Riaño | 42.97 | −5.02 | 101 | 654 | 1100 | 0.95 | Csb | 1 | 2 | 2 | | | 2 | 2 | 0 | 2 | 2 |
| Ricobayo | 41.63 | −5.90 | 100 | 1048 | 684 | 0.27 | Csa | 2 | 4 | 4 | | | 4 | 4 | 0 | 4 | 4 |
| Rosarito | 40.10 | −5.30 | 38 | 84 | 307 | 0.25 | Csa | 13 | 54 | 54 | | | 31 | 43 | 2 | 38 | 41 |
| San Juan | 40.38 | −4.33 | 78 | 148 | 580 | 0.26 | Csa | 1 | 2 | 2 | | | 2 | 2 | 0 | 2 | 2 |
| Sanabria | 42.12 | −6.70 | 88 | 188 | 998 | 0.26 | Csb | 1 | 2 | 2 | | | 2 | 2 | 0 | 2 | 2 |

**Table A1.** *Cont.*

| Name | Position | | Depth | Volume | Elevation | Res. Time | Climate | Visits [a] | Samples | # of Spectroradiometry Samples | | | # of Variables Samples | | | | |
|---|---|---|---|---|---|---|---|---|---|---|---|---|---|---|---|---|---|
| | Lat. | Lon. | m (max.) | ×10⁶ m³ | m.a.s.l. | years | | | | ASD-FR | HandHeld 2 | HR4000 | SDD | CDOM | TSS | Chl_a | PC |
| | d.dd | d.dd | | | | | | | | | | | m | µg/L QSE | mg/L | mg/m³ | mg/m³ |
| Santa Teresa | 40.65 | −5.58 | 59 | 496 | 887 | 0.80 | Csb | 1 | 2 | 2 | | | 2 | 2 | 0 | 2 | 2 |
| Santillana | 40.72 | −3.83 | 40 | 91 | 894 | 0.83 | Csa | 2 | 3 | 3 | | | 3 | 3 | 0 | 3 | 3 |
| Sitjar | 40.01 | −0.23 | 58 | 49 | 160 | 0.37 | Csa | 3 | 6 | | 6 | | 6 | 6 | 6 | 6 | 2 |
| Terradets | 42.05 | 0.88 | 47 | 33 | 372 | 0.04 | Cfa | 1 | 2 | 2 | | | 2 | 1 | 0 | 1 | 0 |
| Tous | 39.13 | −0.65 | 110 | 378 | 135 | 0.28 | Csa | 3 | 9 | | 6 | 3 | 9 | 6 | 9 | 9 | 3 |
| Tremp | 42.22 | 0.97 | 86 | 184 | 501 | 0.18 | Cfa | 1 | 4 | 4 | | | 4 | 4 | 0 | 4 | 0 |
| Ullívarri | 42.93 | −2.58 | 37 | 139 | 547 | 0.30 | Cfb | 2 | 2 | 2 | | | 2 | 2 | 0 | 2 | 2 |
| Valdecañas | 39.80 | −5.43 | 98 | 1446 | 315 | 0.36 | Csa | 3 | 5 | 5 | | | 5 | 5 | 0 | 4 | 4 |
| Valmayor | 40.53 | −4.03 | 60 | 124 | 831 | 3.54 | Csa | 2 | 4 | 4 | | | 4 | 4 | 0 | 4 | 4 |
| Valparaíso | 41.97 | −6.30 | 64 | 145 | 833 | 0.23 | Csb | 2 | 3 | 3 | | | 3 | 3 | 0 | 2 | 2 |
| Valuengo | 38.30 | −6.67 | 32.7 | 10.0 | 297 | 0.08 | Csa | 1 | 2 | 2 | | | 2 | 2 | 0 | 2 | 2 |
| Vega de Jabalón | 38.75 | −3.78 | 25 | 33.5 | 639 | 0.62 | Bsk | 1 | 1 | 1 | | | 1 | 1 | 0 | 1 | 1 |
| Total | | | | | | | | 109 | 296 | 224 | 58 | 14 | 271 | 222 | 92 | 254 | 170 |

## Appendix B. Test of Chl_a and TSS Algorithms Showed in Table 10 Using Image Satellite

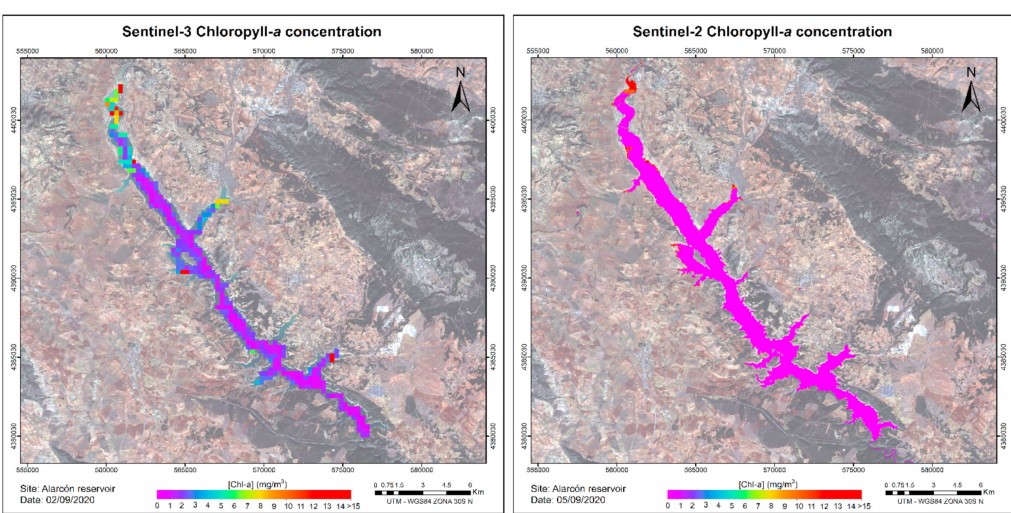

**Figure A1.** Thematic maps of Chl_a concentration for Alarcón reservoir using S3 (left) and S2 (right) images.

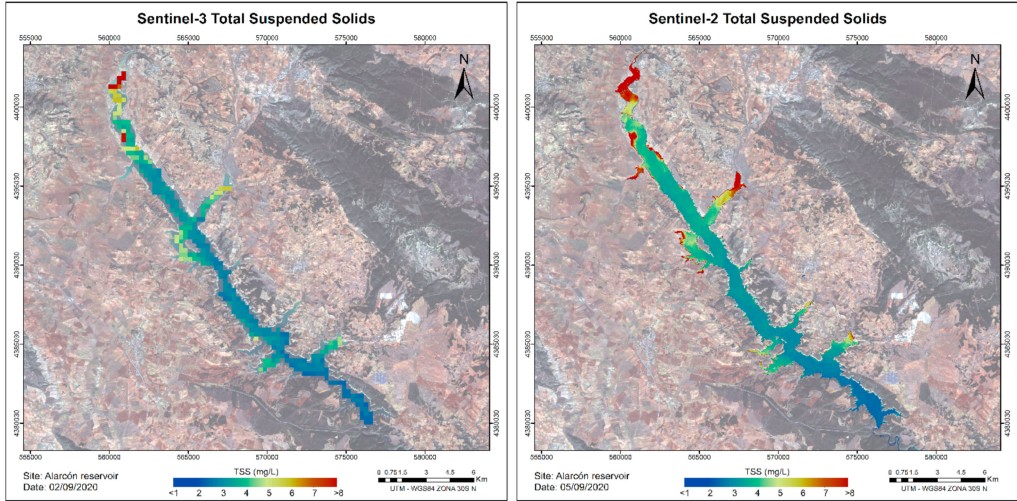

**Figure A2.** Thematic maps of TSS concentration for Alarcón reservoir using S3 (left) and S2 (right) images.

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
