# Peer review of "Validation of Water Quality Monitoring Algorithms for Sentinel-2 and Sentinel-3 in Mediterranean Inland Waters with In Situ Reflectance Data"

_water, doi:10.3390/w13050686_

Round 1

Reviewer 1 Report

Dear Authors,

Thank you for your high quality manuscript. I will recommend it for publication after minor revisions. These are essentially just minor things that I commented on in the attached PDF. But I have one major request: Can you at least include an example image for a Sentinel-2 and Sentinel-3 scene for the parameters TSS and Chl_a (each using the most suitable algorithm). That would improve the quality a lot and illustrate the method well.

Wish you all the best!

Author Response

Thanks for yout time and your constructive comments. Now the manuscript is more consistent.

Reviewer 2 Report

Review

Combined use of Sentinel-2 and Sentinel-3 satellites for Mediterranean inland waters monitoring and management

General Comment

The manuscript entitled ‘Combined use of Sentinel-2 and Sentinel-3 satellites for Mediterranean inland waters monitoring and management’ provides an extensive test trial of different band-based algorithms for the retrieval of optically determined water quality indicators for Spanish lakes. The authors used a large and long-term in situ dataset form numerous lakes and reservoirs, and aligned the respective values with reflectance spectra acquired in the field. Based on the integration of the obtained spectral data into respective band-widths of Sentinel-2 and -3 satellites, different algorithms presented throughout the literature were tested. The best-fit algorithms were identified and are recommended for further use along with real satellite data.

The study includes plenty of interesting data and analyses. However, the context, presentation, and discussion must be considerably improved, before I can recommend publication in ‘Water’.

The title implies, that the work was done on actual satellite data. It took me to read until the discussion section, before I realized, that this is not the case. In this context, it is a major shortcoming of this manuscript that the further steps necessary to transfer the results onto real satellite data are not discussed. I doubt, that the best-fit algorithms based on in situ acquired reflectance spectra will also perform best based on satellite-derived data. Atmospheric correction and detection limits of the sensors are critical and will lead to additional uncertainties, not evenly distributed across across the spectrum. So, I assume that the conclusion of this paper is not as straight-forward as it is presented here.

Further, the discussion is very limited in scope and mainly compares the findings of this study with other very similar studies. I have raised a couple of issues and questions that could be favorably discussed on the background of this dataset See detailed comments).

English language requires considerable revision. There are plenty of grammatically incorrect sentences throughout the text, and I strongly recommend revision by an English native speaker.

Title

The title is misleading as it implies that satellite data is used in the study. It took me to read until the discussion, before I realized that only field measurements of reflectance spectra were used across the respective bandwidths of Sentinel-2 and -3 to perform the analysis. Therefore, the

Abstract

The abstract requires the reader to have high amount of prior knowledge. If the reader never has heard about e.g. Sentinel satellites, it is difficult to understand what has been done. I recommend deepening the explanations a bit. E.g. “Sentinel-2 and -3, two remote sensing satellites of the European Space Agency, are equipped with spectral optical sensors, which are also meant for water body monitoring.”

Further, I am missing clear answers to the following questions: What’s the problem? Why has the study been done? What is the impact of the results?

l.12: Secchi Disk Depth is not an optically active constituent, but a feature of a water column. Please revise.

Introduction

l.30: 21st century

l.41: Revise to e.g.: “These programs require adjustment to regional … “

l.46: In my eyes, costs is not a logistical, but an effort problem. I recommend revising to: “… are also limited by logistics and effort such as access, …”.

l.48: The authors write: “Remote sensing allows a total coverage of water ecosystems with temporal coherent data.” Here I would wish to see a more realistic statement, also including respective limitations, such as shallow waters, stratified water columns, and e.g. cloud cover.

l.52: I recommend “information content” instead of “information contingent”.

l.59: It is redundant to say “Multispectral Instrument (MSI) sensor”. “Multispectral Instrument” is enough.

l.67 It is redundant to say “Ocean and Land Color Instrument (OLCI) sensor”. “Ocean and Land Color Instrument (OLCI)” is enough.

l.103ff: unclear sentence, please revise.

l.116: Shouldn’t this be “for TSS retrieval” instead of “for SDD retrieval”?

l.128: Revise to “… bands closer to the optimal wavelengths for the …“

Material and Methods

l.140: The use of the unit “hm3” seems very unusual to me. I recommend the use of “m3” instead.

l.141: Revise to either: “The climate variability has a great influence on the water quality …”, or “The climate variability greatly influences the water quality …”

l.155: “This work is based on two datasets …”

l.163f: Something’s grammatically wrong in this sentence.

l.171ff: I am wondering, why the samples were taken from these comparatively large depths. From my understanding, the selected sample depths mark the lower boundary of the fraction of the water column where the remote sensing reflectance comes from. So I’d assume that a sample depth in the middle between the surface and this lower boundary would better represent the conditions in that part of the water column represented by the satellite’s signal. The authors should provide a justification for the selection of this sample depth.

Results

l.274: “The cross correlations between variables are given in Table 5.”

l.276: “CDOM shows a high correlation with Chl_a …”

l.278f: Revise to e.g.: “TSS and Chl_a are well correlated (R2 of 0.65). This indicates that in most cases, TSS is mainly composed of phytoplankton and not of suspended minerals.”

l.318ff: I am wondering, whether the statistical measures given in Table 7-10 can be used to compare the quality of the different algorithms tested. I assume that the R2 values represented the level of determination of the transformed values in the regression models (potential, exponential). Were the band ratios, or the concentration values transformed? Further, to me it is not clear, whether transformed or raw concentration values were used to obtain RMSE, RRMSE, and bias statistics. It is crucial to compare the values in the same transformation system to obtain comparable results. This must be clarified.

l.320f: Somethings wrong in this sentence, please revise.

l.346ff: The authors have split the dataset according to different TSS concentration ranges. Before in the text, it is stated that TSS is mostly composed of phytoplankton. It is also possible that different types of TSS require different algorithms. So, I am wondering, whether it is worthwhile to try to split the dataset according to the composition of TSS (if possible?). Maybe there is a couple of samples, where TSS is mainly composed of mineral particles. Even though this might not be possible with the existing data, this issue requires discussion.

Discussion

The discussion mainly focusses on comparing the obtained fit results with other similar studies. Thereby, a lot of the results section is repeated. I am missing several important aspects that could be discussed additionally.

How do the results compare with studies separated in ones that used (1) field reflectance spectra and others that used (2) real satellite data.

The dataset was split in half for calibration and validation. How would it look like, if the dataset was split by different climate types of lakes, as it is mentioned in the study area description? The different lake types will likely have different phytoplankton communities and characteristics of the other optically active features.

What obstacles are there on the way, when trying to transfer the results of this study to real satellite remote sensing data? Atmospheric correction, sun angle, etc.?

l.463ff: First in this paragraph, I realized that no actual satellite data was used in the study. See my comment about the title.

l.561ff: If the same algorithm is used for both Chl_a and PC, this algorithm cannot be used to identify Chl_a or PC independent from each other. Therefore, I recommend deriving only one algorithm for integrated phytoplankton in this case. This is of course based on the assumption of a constant correlation of Cyanobacteria with the rest of phytoplankton. But otherwise, readers get the impression that both variables could be determined independently from each other.

Conclusions

Should be extended according to the issues mentioned for the discussion section.

Author Response

(The authors gave the same response as above.)

Round 2

Reviewer 2 Report

I got the impression that the authors have adequately reacted to my and the other reviewer's comments. The manuscript has been substantially improved and I can now recommend publication in 'Water'

Author Response

Thank you very much. Your comments have helped us to improve our manuscript.